# Merkel Cell Carcinoma: From Pathobiology to Clinical Management

**DOI:** 10.3390/biology10121293

**Published:** 2021-12-08

**Authors:** Peerzada Umar Farooq Baba, Zubaida Rasool, Ishrat Younas Khan, Clay J. Cockerell, Richard Wang, Martin Kassir, Henner Stege, Stephan Grabbe, Mohamad Goldust

**Affiliations:** 1Department of Plastic, Reconstructive Microsurgery, Sher-i-Kashmir Institute of Medical Sciences, SKIMS, Srinagar 190011, India; drumar397@gmail.com; 2Department of Pathology, Sher-i-Kashmir Institute of Medical Sciences, SKIMS, Srinagar 190011, India; drzubaida@rediffmail.com (Z.R.); Ishratashfaq3115@gmail.com (I.Y.K.); 3Departments of Dermatology and Pathology, UT Southwestern Medical Center, Dallas, TX 75390, USA; ccockerell@dermpath.com; 4Cockerell Dermatopathology, Dallas, TX 75390, USA; 5Department of Dermatology, UT Southwestern Medical Center, Dallas, TX 75390, USA; Richard.Wang@UTSouthwestern.edu; 6Founding Director, Worldwide Laser Institute, Dallas, TX 75390, USA; theskindoctor@aol.com; 7Department of Dermatology, University Medical Center of the Johannes Gutenberg University, 55122 Mainz, Germany; henner.stege@unimedizin-mainz.de (H.S.); stephan.grabbe@unimedizin-mainz.de (S.G.)

**Keywords:** Merkel cell carcinoma, polyomavirus, sentinel lymph node, UV-radiation

## Abstract

**Simple Summary:**

Merkel cell carcinoma (MCC) is an uncommon type of skin cancer that carries a poor prognosis. It is seen predominantly in old age in sun-exposed body parts. Racial and geographical differences are seen in its occurrence. Viral infection and radiation exposure are the two leading factors implicated in its causation. Small, firm to hard nodule (usually in sun-exposed areas), red with a history of a rapid increase in size is a common personation of the disease. Other body parts such as upper limbs, trunk, and even lower limbs may be also involved. The disease is diagnosed by taking a tissue sample (biopsy) for examination, and other radiological investigations are needed to reach a proper diagnosis with the staging of the disease. There are various treatment options including surgery, radiotherapy, and chemotherapy. Surgery is the primary treatment option though some patients may not be the candidates for operation where other treatment options come into play.

**Abstract:**

Merkel cell carcinoma (MCC) is an infrequent, rapidly growing skin neoplasm that carries a greater probability of regional lymph node involvement, and a grim prognosis in advanced cases. While it is seen predominantly in old age in sun-exposed body parts, the prevalence varies among different races and geographical regions. Merkel cell polyomavirus and UV radiation-induced mutations contribute to its etiopathogenesis. The clinical presentation of MCC lacks pathognomonic features and is rarely considered highly at the time of presentation. Histopathological examination frequently reveals hyperchromatic nuclei with high mitotic activity, but immunohistochemistry is required to confirm the diagnosis. Sentinel lymph node biopsy (SLNB) and imaging are advised for effective staging of the disease. Multimodal management including surgery, radiation therapy, and/or immunotherapy are deployed. Traditional cytotoxic chemotherapies may result in an initial response, but do not result in a significant survival benefit. Checkpoint inhibitors have dramatically improved the prognosis of patients with metastatic MCC, and are recommended first-line in advanced cases. There is a need for well-tolerated agents with good safety profiles in patients who have failed immunotherapies.

## 1. Introduction

Merkel cell carcinoma (MCC) is an infrequent, rapidly growing skin neoplasm initially described by Cyril Toker in the year 1972. The “cell of origin” of MCC is yet to be fully substantiated. Based on the similarities in biology and morphology of the ‘Merkel cells’ (MC), which are epidermal cells with both epithelial and neuroendocrine attributes and MCC, it has been postulated to arise from the MCs and is thus occasionally described as neuroendocrine carcinoma of the skin [1,2,3,4,5,6,7,8]. Epidermal stem cells, dermal fibroblasts, B cell precursors, and neuronal precursor cells have all been proposed as possible cells of origin for MCC [8]. Though MCC can involve any anatomical part/region in the human body, it occurs predominantly in the solar unshielded body parts in older males, with numerous studies substantiating the effects of long-term ultra-violet radiation exposure [3,4]. Most importantly, MCC is strongly associated with immunosuppression, both iatrogenic and age-related [4,5,6,9,10,11,12,13,14]. It is frequently associated with regional lymphadenopathy and disseminated disease can spread to many parts of the body. The high mortality rate renders it twice as deadly as melanoma [3,4,15].

## 2. Epidemiology

The etiopathogenesis of MCC involves an intricate mutual interaction among various intrinsic and extrinsic factors and influences. Although the specific cause is yet to be determined, two aetiologies that have been widely discussed and postulated include viral oncogenesis and radiation [3,6,9,15]. The identification of MC polyomavirus (MCPyV) in most tumor specimens by Feng et al. proved to be a landmark study in our understanding of its biology. Furthermore, a study by Touzé A et al. in 2009 confirmed this association of MCPyV with MCC [16,17]. While the precise mechanisms by which MCPyV promotes tumorigenesis are debatable, the expression of MCPyV early region genes plays a critical role in oncogenesis. UV radiation appears to synergize with MCPyV to promote tumorigenesis, and it has been suggested that mutations to the viral genome promote viral integration and tumorigenesis [5]. Notably, psoriasis patients on phototherapy treatment are 100 times more predisposed to develop MCC, and correlations between MCC and radiation index and decreased skin pigmentation have been noted [15,16]. Accordingly, the predominant, but not exclusive, sites of involvement are exposed parts including the head and uncovered areas of the upper limbs (Figure 1) [17,18,19]. A compromised immune system is also a well-established risk factor for MCC; iatrogenic immune suppression, lymphoproliferative disorders such as chronic lymphocytic leukemia, autoimmune disorders, organ transplant candidates, and acquired immunodeficiency syndrome are known to increase the likelihood of MCC [18,19,20]. The high prevalence of MCC in older patients suggests that immunosenescence may contribute as well [5,6].

MCC occurs less frequently than melanoma, squamous, or basal cell cancers [16]. Although the incidence appears to be increasing in the previous few decades, the rarity of the tumor hinders the assessment of the true incidence. Improved diagnostic techniques, a growing elderly population, and an increase in the number of immune-compromised patients may all be contributing to the rise in incidence [2,21]. Incidence rates vary across different parts of the world. As is true for other UV radiation-induced cutaneous malignancies, Australia tops the list with the incidence of 16 cases per million population [15]. Notably, the incidence of MC Polyoma virus-positive cases is low in Australia [11]. The age-adjusted annual incidence from the United States, Europe, and Australasia lies between 0.1 and 1.6 per 100,000, with the number of cases reported from Asia the lowest [3,15]. In the United States, about 1600 new cases are detected every year [5,6,9].

The frequency of MCC differs among varied ethnic clusters and regional zones-prevalence being 25 times higher in the white population than dark skin types. The classic patient is an elderly white man aged above 70 years with a history of prolonged sun exposure [15,22]. The age-adjusted incidence rate is 8 times lower in the black populace [5,9,15]. Most studies confirm that males outnumber females; with some demonstrating a gender ratio of up to 2:1. However, though some small studies have demonstrated either not predilection or a slight female preponderance [17,22]. Females tend to be older than males at presentation (76 years vs. 71 years) [13]. The lesions are mostly encountered in the head with the trunk the favored location in younger patients and lower limbs in African Americans [5].

Following past studies, the mean age at diagnosis in most recent studies is reported to be 74.9 years. It is extremely rare in children and young people, being 24 times more common after 65 years of age and only 5% diagnosed before 50 years. Two studies (one from New Zealand and another US-based) demonstrated an enhancement in the age-specific incidence in old age group patients when compared with a younger group [6,22].

## 3. Clinical Presentation

The clinical manifestations of MCC are protean and are rarely suspected at the time of presentation [5,15,16]. One common presentation is a single, firm to hard, rounded, red to violaceous nodule or tumors that are non-tender and rapidly growing [22,23]. (Figure 1, Figure 2 and Figure 3).

The tumors are observed more commonly on the left side than on the right. Such an observation has not revealed any correlation with the radiation exposure, age, sex, site predilection, or MCPyV status [24]. The explanation for such laterality of the lesions continues to remain indeterminate. Gambichler T et al. reported the left-sided predominance of the tumors in the ratio of 8:I (torso), 1.8:1 (head and lower limbs) and 1.2:1 (upper limbs), et [24] Lesion tends to grow as hemispheres towards the outside (giving typical dome shape) or in an iceberg fashion to inside [10]. Rapid growth with ulceration can be observed.Multiple morphologic features including papules, plaques, cysts, pruritic growths, stalked neoplasms, deep cutaneous masses, and lesions with telangiectasia, have all have been reported in the literature [15,16]. The majority (70–90%) of these lesions have an affinity for the head/neck or upper extremities (Figure 1). Head and neck lesions are typically smaller than those from other regions [16]. Additional diagnoses in the differential may include basal/squamous cell carcinoma (BCC/SCC), lymphoma cutis, pyogenic granuloma, sarcoma, epidermal inclusion cyst, dermatofibroma, adipose tumor, melanoma (amelanotic type), or metastatic lesions, and should be differentiated from them through biopsy [16,23]. Carneiro et al. reported concomitant malignancy in 18.7% of patients in the form of BCC, SCC, and sebaceous carcinoma [9]. Heath et al. have devised an initialism (AEIOU) to elucidate the typical manifestations of MCC: asymptomatic, rapid expansion, immune suppression, old age (>50), and ultraviolet radiation exposure. In his study, ≥3 of these traits were found in about 89% of the patients [14,15]. Mucosal MCC’s (e.g., oral/genital), although rare, have been reported to be more aggressive than the cutaneous ones [15,16]. At presentation, localized cutaneous disease is documented in about two-thirds of patients; however, cases with the nodal disease at presentation are not infrequent. The US National Cancer Database reveals two-thirds of patients presenting with localized tumor followed by draining lymph node basin adenopathy in 27%, while 7% exhibit metastasis to remote areas. In contrast, the SEER Program reported local disease in <50% cases with the same percentage of metastasis. In 14% of patients, despite the documented metastasis, the primary tumor cannot be identified [13,18]. It is to be emphasized that ‘‘visceral MCC’’ without any skin manifestation is currently regarded as a distinct subtype of neuroendocrine carcinoma [5,13,15]. This should be distinguished from nodal only MCC, which is thought to be MCC with spontaneous resolution of the cutaneous lesion. Importantly, MCC in situ is extremely rare. Basically, these are dermal tumors, as already described, with the involvement of epidermis alone extremely unusual; only a very few cases of such stage 0 disease are reported in the literature [25].

## 4. Etiopathogenesis

The pathogenesis of MCC is partially known. In addition, the cell of genesis for MCC also remains unclear and is currently debated [6,10]. Merkel cells, named as ‘‘touch cells,’’ when originally reported by Frederich Merkel long back in 1876, are seen mainly in the basal layer of the epidermis. They are physiologically implicated in tactile response and somatostatin production and have a presumed role in other endocrine functions. This explains the historical designation of ‘‘neuroendocrine’’ for MCC tumors [5,6,9,10]. While MCC’s are broadly classified into 3 histological subtypes—trabecular, intermediate, and small cell—there is not yet any substantiation for the clinical utility of such classifications [2,3]. The trabecular type (least common) is well-differentiated and is commonly observed in mixed tumors. The intermediate type is most common (observed in ≥50% of tumors) with high mitotic activity. The small-cell type is poorly/undifferentiated and appears similar to small cell carcinoma of other locations, especially the lung [22,23]. Pathologically, MCC is typically located in the dermis and occasionally in the sub-dermis with sparing of the epidermis in general [25]. At times, the monotonous appearance of the “small round cells” in the dermal layer and subdermal adipose tissue, makes MCC indistinguishable from malignant lymphomas with immunohistochemical stains [26,27]. A range of trabecular, insular, or diffuse growth patterns may be seen, with either one or all patterns coexisting within an individual tumor. The dermis is often filled, with a very narrow area of uninvolved papillary dermis set apart from the epidermis by a slender zone called the ‘Grenz zone’. The cytoplasm can be visualized as a thin acidophilic rim and is amphophilic to eosinophilic. The following are some histopathologic pictures of MCC (Figure 4, Figure 5 and Figure 6).

The nuclei are relatively uniform and grayish being round and vesicular, having a typical fine granular “pepper and salt” appearing chromatin motif and multiple nucleoli; hyperchromatic nuclei and high mitotic activity are characteristic [4]. Mitotic figures are abundant and fragmented nuclei are plentiful. Necrosis may be present, occasionally associated with hematoxylin staining of vessel walls and fibrous septa known as the “Azzopardi phenomenon” [28]. These histological features need to be distinguished from other so-called ‘small round cell tumor’ groups as well as from melanoma. Accurate diagnostic discrimination of these entities is critically important for predicting long-term prognosis determining the appropriate treatment. As no histologic features can reliably distinguish MCC from other small round cell tumors, immunohistochemical stains prove useful in excluding these differentials [19]. Immunohistochemically, MCC shows positivity for low molecular weight keratin and neurofilaments, which appear as dot-like condensations of filaments. In particular, it is strongly positive for keratin type 20 with distinct dots concentrated around the nucleus [9,29,30]. CK20 and neurofilament positivity and Thyroid transcription factor 1 (TTF-1) negativity are important to designate MCC from histologically similar neoplasms. CK20 staining in MCC is often dot-like but a decoration of the cytoskeleton is also possible [31,32,33]. In rare cases, MCC may stain for CK7, while being negative for CK20 [34]; CK19 has its diagnostic significance for CK20-negative MCC [35]. Consequently, tumors that are positive for CK20 along with TTF-1 negativity indicate MCC [22,23]. Immunohistochemical demonstration of neuron-specific enolase, intense staining for keratin (CK20), and absence of S100 protein, leukocyte common antigen, vimentin, and HMB45 are enough to confirm the diagnosis [35,36] (Table 1). In addition to the above-mentioned markers, cases of MCC manifest reactivity with chromogranin, synaptophysin, vasoactive intestinal peptide, substance P, pancreatic polypeptide, calcitonin, adrenocorticotropic hormone, somatostatin, other peptide hormones, PAX-5, TdT, glypican-3, and CD117 [37,38,39,40,41,42,43,44].

## 5. Diagnostic Evaluation/Staging

Diagnostic evaluation starts with proper history and clinical examination. Quite atypical and diverse gross manifestations constitute diagnostic challenges even for experienced clinicians. Sentinel lymph node biopsy (SLNB) is advocated in most cases for definitive disease staging since 33% of the patients with clinical evidence of primary disease harbor occult disease in draining lymph node basin [15,18].

The guidelines from the US National Comprehensive Cancer Network (NCCN) for diagnostic evaluation include: differentiation from nodal only MCC, which is thought to be MCC with spontaneous resolution of the cutaneous lesion, imaging modalities such as magnetic resonance imaging (MRI), computed tomography (CT), or positron emission tomography (PET) can be performed as clinically indicated; and when possible, individual cases ought to be discussed in a multi-facility tumor board of the institution [14,16,22]. According to the updated American Joint Committee on Cancer (AGCC) staging system for MCC (Table 2), according to life expectancy, patients are categorized into different groups. This staging system blends clinical and pathological elements of the disease to facilitate appropriate treatment strategies. Additionally, it regrouped pathologically confirmed nodal disease with unconfirmed primary lesion into stage IIIA, highlighting the improved clinical outcomes connected to it [13,14,15].

## 6. Prognostic Factors and Disease Progression

Usually, MCC does not have a good prognosis, especially when the initial presentation is that of metastatic disease. Despite substantial insights into the tumor biology of MCC, survival rates continue to be poor for advanced disease. Until the advent of checkpoint inhibitor treatment, the main therapeutic option for advanced MCC has been chemotherapy, where there have been seen initially good responses though without any significant survival benefit. With the advent of checkpoint inhibitors, progression-free survival (PFS) and overall survival (OS) are refined dramatically in advanced disease. A single-arm trial with Pembrolizumab/Avelumab leads to the approval of the use of checkpoint inhibitors by the FDA and EMA for the treatment of advanced MCC [45,46]. The estimated overall five-year survival for local disease is 51% whereas it stands at 14% for metastatic disease. A study from the Netherlands reported 1-, 5-, and 10-year relative survival of 85%, 62%, and 47%, respectively [47]. However, the 5-year survival rate in a study by Agelli and Clegg was reported to be 75%, 59%, and 25% for local, regional, and distant disease, respectively [7]. A study by Tam et al. revealed that females have a survival advantage over men [48]. Disease-associated mortality is rated at 33–46%, which markedly surpasses that seen in melanoma patients, assigning it as the second leading cause of skin cancer death [15,22]. Recurrence is usually encountered during the initial 2 years of disease. Frohm et al. observed a mean of 10.7 months for the first recurrence to appear with a range of 3.5 to 23.3 months [49]. MCV positivity is known to predict favorable outcomes when compared to similar tumors that are MCPyV negative [4,15,19]. A subset of MCPyV-positive patients produce MCPyV-specific antibodies, which also serve as a favorable prognostic factor. The presence of antibodies against an MCPyV T antigen, but not capsid protein antibodies, predicts recurrent or progressive disease if observed 1-year post tumor excision [50]. Furthermore, CD8 T cell counts in the neoplasm are considered to predict survival; a study by Feldmeyer et al. proposed tumoral immune infiltrates as a strong prognostic indicator [51]. Finally, some histologic and immunohistochemical features have been reported to predict disease progression. Specifically, rapid mitosis, small cell morphology, the paucity of inflammatory infiltrates, lymphovascular involvement, p53 positivity, p63 positivity, raised mast cell-cell counts, and dense tumoral/peritumoral vascularity all have been proposed to be negative predictors, with p63 exhibiting the greatest prognostic value [52,53,54,55,56,57]. The strongest negative prognostic indicators include male gender, advanced age, comorbid conditions, immunocompromised state, head and neck localization, tumor of 2 cm or more, advanced stage (clinically presence of lymph nodes, distant metastases), and positive resection margins [5,9]. It is worth noting that the time interval between the presentation and initiation of treatment is crucial to prognosticate the disease progression [4]. The prognostic utility of tumor thickness continues to be controversial. While some authors indicated that a nodal disease with a poor prognosis is related to tumor thickness, others have refuted it [9].

Nodal MCC without any evidence of a primary tumor is associated with improved survival, perhaps due to an active immune-mediated clearance of the cutaneous tumor before its presentation. A potential immunological clearance of the carcinoma is seconded by several reports of spontaneous regression of the tumor including distant metastatic ones [15,16]. Advancement to metastatic disease is classically noticed within the first 3 years of diagnosis. Distant metastasis has most commonly been observed in the liver, lung, brain, bones, and remote skin. Less frequent sites of advanced disease include gastrointestinal epithelia, pancreas, parotid gland, pleura, cardiac tissue, bone marrow, prostate, testicles, and urinary bladder [13,16,18].

## 7. Treatment Options

Depending upon the clinical stage, comorbidities, and performance status of the patient, multimodality management protocols consisting of extirpation of the primary lesion, regional lymphadenectomy with radiotherapy/chemotherapy is being employed as the optimal treatment. However, at present, there is debate on the most efficacious treatment scheme for the advanced disease; moreover, it should be distinguished from nodal only MCC, which is thought to be MCC with spontaneous resolution of the cutaneous lesion [9,10,11].

Operative management is the gold standard for localized or locoregional disease. Wide excision with a margin of 1–2 cm (1 cm for <2 cm tumor size and 2 cm for >2 cm ones) is usually recommended [15,20]. Surgery resulting in positive margins is a candidate for re-excision. In sites where tissue sparing is critical, Mohs microsurgery may be an option, but lack of unanimity precludes its utility. Lymph node dissection followed by radiation is the standard treatment for Stage III disease. SLNB is typically recommended at the time of the primary surgery for the cases without nodes on clinical examination. For those with positive SLNB, elective lymph node dissection (ELND) with radiation is the norm [13,20].

Radiotherapy (RT) is utilized both as an adjuvant to surgery and as a monotherapy in those in whom surgery is not conducive or suitable, e.g., an elderly patient with a plethora of comorbidities with high risk for anesthesia/surgery. Most frequently, it is employed for the locoregional control of the disease. Most literature suggests that recurrence rates are considerably reduced with adjuvant RT, which has better results than adjuvant chemotherapy [58,59,60,61]. The cumulative radiation dosage regimen for the primary site ranges between 50 and 56 Gy for margin negative surgery with dosage enhancement up to 60 Gy for positive histological margins and a maximum of 66 Gy for gross disease (standard fractions ranging from 1.8 to 2 Gy per session) (Table 3) [58].

Cook et al. stated that postoperative single-fraction radiotherapy is a promising option to conventionally fractionated postoperative radiotherapy to treat the primary lesion in localized head/neck Merkel cell tumors, particularly in elderly and frail patients (not candidates for surgery), with potential control of disease without noticeable toxicity [62]. A retrospective study in the Moffitt Cancer Center revealed that postsurgical radiation improves disease control as well as disease-specific survival in cases having positive nodal disease, but without any effect in patients with no nodal involvement. The international guidelines recommend that adjuvant radiation should be initiated within 4–6 weeks of surgery for all stages of the disease (I–III). (NCCN 2021) [63]. In general, for large node-negative tumors, consensus panels and international guidelines agree that adjuvant RT following operative extirpation improves the local control in the tumor bed, whereas controversy continues for its use in stage III disease, except for those with multiple lymph nodes with confirmed extracapsular extension [59]. The National Cancer Database analysis [64] has validated that the adjuvant RT improves overall survival in stage I and II patients, but not in stage III disease. This was further confirmed in a multicenter retrospective study by Servy et al. [65]. which concluded that survival in stage III depends on the presence/absence of subclinical distant metastases rather than the inclusion of specific adjuvant therapy [59].

Radiation monotherapy has most commonly been reported in head and neck tumors [59]. Ott et al. [66] reported a prolonged remission in four such patients. Similarly, Lawenda et al. concluded that radiation to the primary tumor bed, either definitive or as an adjuvant following resection, could produce a local control rate exceeding 90% [61]. Mortier et al. demonstrated the validity of managing inoperable cases by RT alone, with the results at least equal to ones managed by surgery and radiotherapy [60]. In addition, good results have been witnessed with single-fraction radiation therapy in metastatic tumors. It reduced the tumor burden with durable palliation and limited side effects [12,13,14,15,16]. Palliation of bone metastases and other extracranial metastatic sites can be achieved with 10 fractions of 3 Gy each and/or a single 8 Gy fraction. The University of Washington reported complete responses in 45% and local control in 77% cases of metastatic tumors when treated with a single 8 Gy fraction [58].

Though MCC is generally regarded as a chemotherapy-responsive tumor, results are hardly ever durable. There are no documented favorable results and clear benefits with the use of chemotherapeutic regimens. It is typically reserved for palliative therapy in stage IV disease where other treatments have failed. Cytotoxic chemotherapies tend to increase morbidity and reduce the quality of life with their associated immunosuppression and toxicity. Resistance to chemotherapy on recurrence is another drawback. Usual drug regimens comprise cisplatin/carboplatin with etoposide/topotecan [12,13,14].

Immunotherapy has materialized as a promising management tool for advanced MCC. Various research papers have corroborated the association of immune status to clinical outcomes, opening the way for beneficial effects of cell-mediated immunity [67]. Three antibodies targeting the PD-1 axis have been extensively studied, with all exhibiting significant response rates and good response durability (Table 4).

Among these, pembrolizumab (anti–PD-1) was the first immune inhibitor demonstrating tumor regression [67]. A National Cancer Institute-sponsored clinical trial studied the effect of pembrolizumab in 25 cases of advanced disease, showing an ORR of 56% including a 16% complete response rate (CRR) and an estimated progression-free survival (PFS) of 67% (at 6 months) with the response duration ranging from 2.2 to 9.7 months. ORR did not differ significantly in virus-negative and virus-positive tumors (62% vs. 44%). Based on these observations, pembrolizumab has been recommended for disseminated MCC (National Comprehensive Cancer Network (NCCN) guidelines). Nghiem et al., in another multicenter phase II trial (Cancer Immunotherapy Trials Network-09) involving 50 cases of advanced MCC, treated with pembrolizumab (2 mg/kg every 3 weeks) for up to 2 years, showed an objective response rate (ORR) of 56% (complete response 24%; partial response 32%; 95% CI, 41.3% to 70.0%), with ORRs of 59% in virus-positive and 53% in virus-negative tumors. The 24-month PFS rate was 48.3%, and the median PFS time was 16.8 months (95% CI, 4.6 months to not estimable) with the 24-month OS rate of 68.7% [46]. Nivolumab is another monoclonal PD-1 antibody with clinical efficacy in advanced tumors. Twenty-five patients—treatment-naïve and previously treated (36% and 64%, respectively), MCPyV-positive/negative, advanced MCC—were enrolled and treated with nivolumab 240 mg every 2 weeks with a median follow-up of 51 weeks. Investigators observed an ORR of 64% and PFS of 82% at 3 months. Avelumab, a PD-L1 monoclonal antibody, was approved by the FDA in 2017. Approval was based on data from an open-label, single-arm, multi-center phase 2 clinical trial. In this study, 88 patients with advanced stage received avelumab 10 mg/kg every 2 weeks. This trial demonstrated an ORR of 33%, with a CRR of 11% and PFS of 40% (at 24 weeks) with the estimated PFS of 30% at 12 months. Unlike pembrolizumab, ORR was not significantly different in virus-positive/negative tumors (26% and 35%) [45,68]. The results of the above-described trials led to the inclusion of avelumab, pembrolizumab, and nivolumab in the NCCN guidelines (in January 2018) as the preferred treatment options for disseminated disease (Table 4) [67,69,70,71]. With the advent of ICIs, for the first time, long-lasting response durability was observed in patients with advanced MCC. Even if adverse events due to checkpoint inhibitors are known, re-induction is a treatment option in case of tumor progression. Stefe et al., in a retrospective multicenter study, evaluated patients with re-induction of anti-PD-1/anti-PD-L1 therapy; following a mean treatment-free period of 9.5 months (3–18 months), re-induction with ICI therapy achieved an objective response in five of eight patients (62.5%) upon re-induction, while no response could be observed in the rest. Notably, adverse events compelling halt of treatment were not observed during reinduction [70].

A variety of agents belonging to other classes have been tried in some cases. TNF-a, interferon-a (IFN-a), anti-CD56 antibodies, or vaccines (IL-12 gene using vaccine) have all been reported beneficial in case reports or small series. Though case series/reports regarding the use of tyrosine kinase inhibitors (Imatinib, cabozantinib, and pazopanib), and somatostatin analogs (lanreotide, octreotide, and pasireotide) do exist, clinical trials supporting their use have not yet been conducted, therefore, these agents are not included in management protocols [10,12,20]. In advanced tumors, various intralesional immunotherapies were tried. Intralesional therapy with tumor necrosis factor, class I interferon, and talimogene laherparepvec (T-vec) have revealed promising results. Intralesional TLR4 agonist and IL-12 administration (through electroporation) resulted in persistent objective responses in individuals with advanced disease in early-phase trials. Currently, trials are in place for T-vec alone or incorporation with radiation or nivolumab, and TTI-621 (anti-CD47). Furthermore, a phase I/II trial of intralesional TLR7/8 agonist plus a modified IL-2 formulation has been initiated. Though none have been approved yet, still, one more triple-combination study of tremelimumab (an anti–CTLA-4 antibody), durvalumab (an anti–PD-L1 antibody), and intratumoral TLR3 agonist poly-ICLC is underway [71].

## 8. Conclusions

Although substantial insight into the pathogenesis and advancements in the treatment protocols have been achieved in the recent past, mortality rates and projections have improved little in advanced-stage disease. The tumor stage at the time of presentation is still the most dependable prognostic predictor. Multiple management modalities must be tailored to minimize morbidity while maximizing the chances of survival. The continued development of well-tolerated immuno-therapies with acceptable side effects profiles may improve treatment options for patients with recurrent or persistent disease. Despite the increasing incidence, few institutions manage enough patients that they can effectively analyze the data regarding the efficacy and safety of different treatment options on their own. Hence, prospective, multicenter, randomized case-control studies should be considered to continue to refine and standardize treatment guidelines.

## Figures and Tables

**Figure 1 biology-10-01293-f001:**
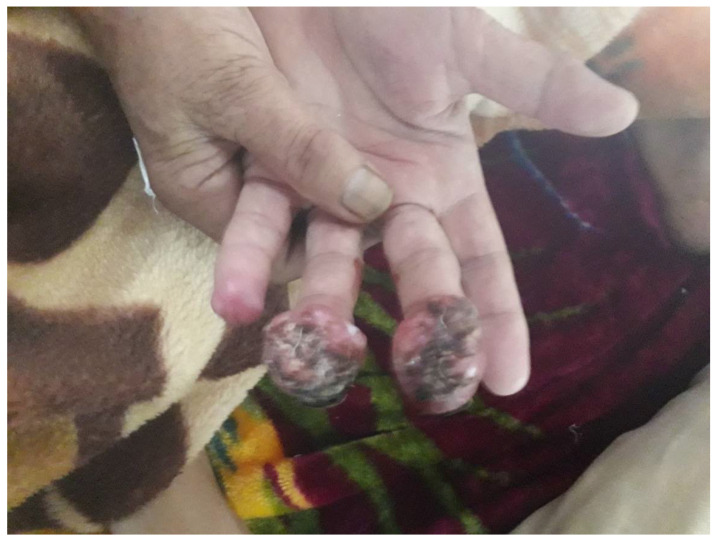
Merkel cell carcinoma involving the middle and ring fingers of the left hand of a 76-year-old female.

**Figure 2 biology-10-01293-f002:**
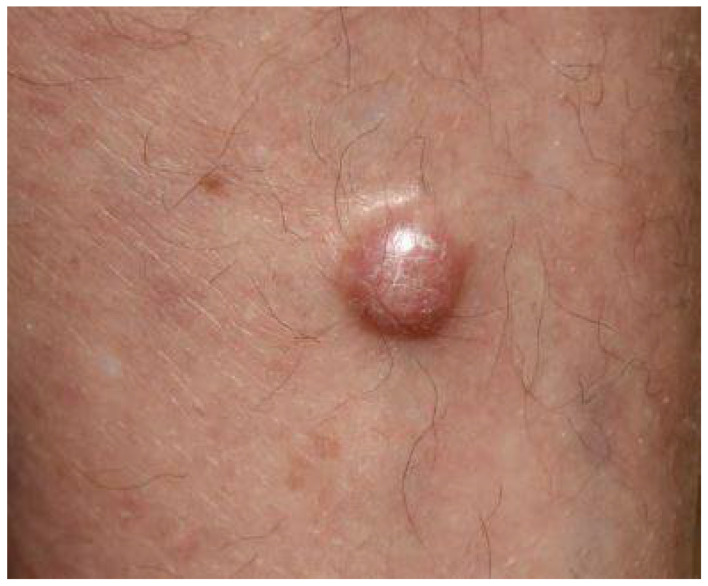
Nodular variant of Merkel cell carcinoma.

**Figure 3 biology-10-01293-f003:**
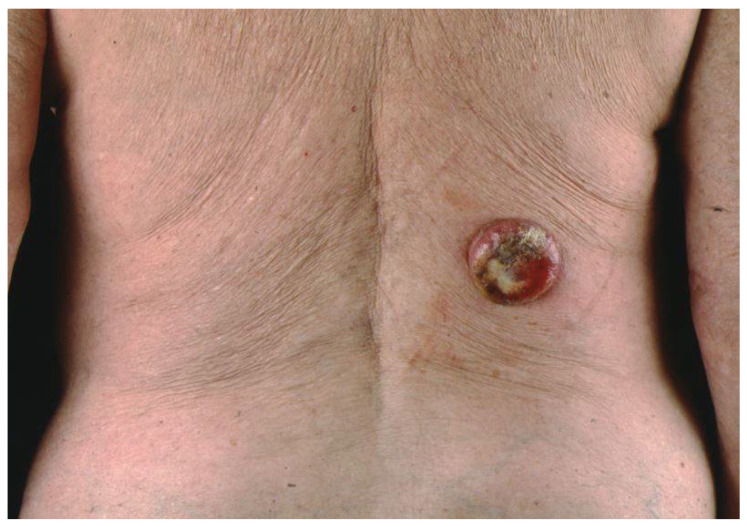
Merkel cell carcinoma involving right lumbar area.

**Figure 4 biology-10-01293-f004:**
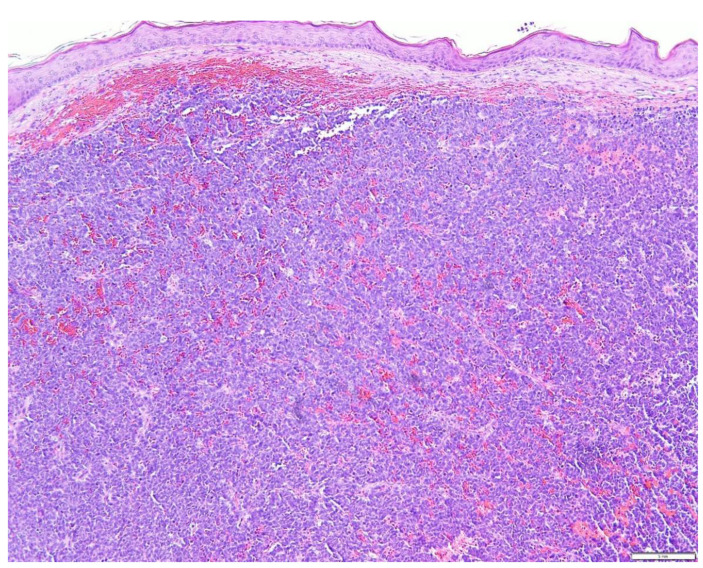
Beneath a normal appearing epidermis, a dense diffuse population of basophilic cells arranged in a nodular pattern replaces the dermal collagen (H&E × 10).

**Figure 5 biology-10-01293-f005:**
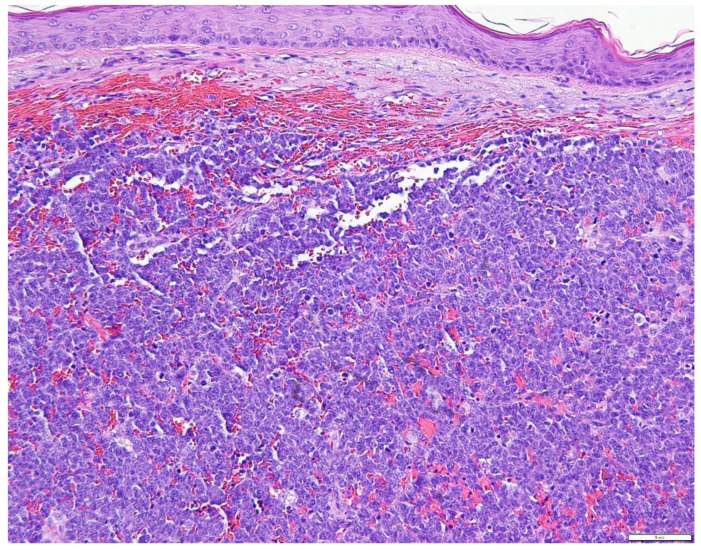
Higher magnification shows a population of pleomorphic round and oval shaped basophilic cells with relatively indistinct borders interspersed with occasional lymphocytes and numerous erythrocytes and mitotic figures (H&E × 20).

**Figure 6 biology-10-01293-f006:**
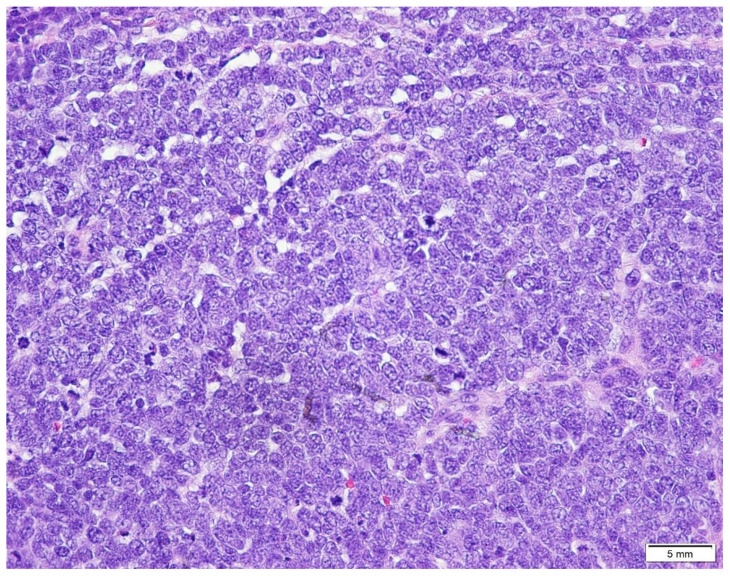
High power magnification reveals nuclear molding and many mitotic figures within the dense population of round and oval shaped basophilic malignant cells with scant cytoplasm and indistinct borders (H&E × 40).

**Table 1 biology-10-01293-t001:** Immunohistological markers of MCC and other differential diagnoses [8,37,38].

Marker	MCC	Lymphoma	Melanoma	SCLC
Cytokeratin 20 (CK 20)	+	−	−	−
Cytokeratin7 (CK 7)	−	−	−	+
Chromogranin A	+/−	−	−	+/−
HBM45	−	−	+	−
Huntingtin-interacting protein 1 (HIP1)	+	+/−	−	−
Melan-A/MART-1	−	−	+	−
Leucocyte common antigen (LCA)	−	+	−	
S100B	−	−	+	−
Thyroid transcription factor 1 (TTF-1)	−	−	−	+
Neuron-specific enolase	+	−	−	+/−
Vimentin	−	+	+	−

**Table 2 biology-10-01293-t002:** Merkel cell carcinoma staging system (American Joint Committee on Cancer; 8th edition) [5].

Stage	Primary Tumor	Lymph Node Status	Distant Metastasis
0	In situ	No nodes	No
I Clinical	Tumor ≤ 2 cm in maximum dimension	No nodes on clinical examination (histopathological examination not performed)	No
I Pathological	Tumor ≤ 2 cm in maximum dimension	Nodes negative by histopathologic examination	No
IIA Clinical	Tumor > 2 cm	No nodes on clinical examination (histopathological examination not performed)	No
IIA Pathological	Tumor > 2 cm	Nodes negative by histopathologic examination	No
IIB Clinical	Primary tumor involves fascia, muscle, bone, or cartilage	Clinically absent nodes (histopathology not performed)	No
IIB Pathological	Primary tumor involves fascia, muscle, bone, or cartilage	Histopathology demonstrates negative nodes	No
III Clinical	Tumor of any size/depth	Clinically nodes are present (histopathology not performed)	No
IIIA Pathological	Tumor of any size/depth	No nodes present on clinical examination, but nodes positive by pathological examination	No
	Unknown primary	Nodes present on clinical examination and confirmed by histopathological examination	No
IIIB Pathological	Tumor of any size/depth	Clinically nodes are present which are confirmed by histopathology or in-transit lesions	No
IV Clinical	Any	Present/absent	Clinically metastasis is present
IV Pathological	Any	Present/absent	Histopathological confirmation of metastasis

**Table 3 biology-10-01293-t003:** Radiotherapy dose guidelines for Merkel cell carcinoma [67].

Site	Clinical Scenario	Recommended Dose
Primary	Surgical resection with wide margins (e.g., 1–2 cm) Small tumor size (<1–2 cm)	Consider observation
	Surgical resection with negative margins	50–56 Gy
	Surgical resection with histologically positive margins	56–60 Gy
	Surgical resection with gross positive margins	60–66 Gy
	No resection performed; as definitive therapy	60–66 Gy
Nodal basin	Node-negative (clinical examination); SLNB negative	Observation
	Node-negative (clinical examination); SLNB/lymph node dissection not performed	46–50 Gy
	Node-negative (clinical examination) SLNB positive, lymph node dissection not performed	50–56 Gy
	Lymph node dissection with extracapsular extension; multiple positive Lymph nodes	50–60 Gy
	Node positive (clinical examination); SLNB or lymph node dissection not performed	60–66 Gy

Abbreviation: SLNB, sentinel lymph node basin.

**Table 4 biology-10-01293-t004:** Immune checkpoint inhibitor trials in advanced Merkel cell carcinoma [67,69,71,72].

	Avelumab	Pembrolizumab	Nivolumab	Ipilimumab
Mechanism of action	Anti-PD-L1	Anti-PD-1	Anti-PD-1	Anti-CTLA-4
Dose and schedule of administration	10 mg/kg IV q2 week	2 mg/kg q3 week	240 mg IV q2 week	3 mg/kg IV q3 weeks × 4 doses
ORR in chemotherapy naïve MCC	69% (*n* = 16)	56% (*n* = 26)	71% (*n* = 14)	40% (*n* = 5)
ORR in chemotherapy treated/second line MCC	33% (*n* = 88)	N/A	63% (*n* = 8)	
The median time of response	6.1 weeks	12 weeks	2 months	

Abbreviations: IV = Intravenous; ORR = Objective response rate; N/A = Not available.

## Data Availability

Not applicable.

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
