# Peer review of "Merkel Cell Carcinoma: From Pathobiology to Clinical Management"

_biology, 2021, doi:10.3390/biology10121293_

Round 1

Reviewer 1 Report

This paper is a comprehensive review article of Merkel cell carcinoma (MCC) summarizing many biological, clinical and therapeutic aspects of MCC. It is  done properly and includes many citations( 70). However, there are some short-comings :

  1. MCC is typically a subcutaneous red-violaceous nodule without ulceration. Ulceration is typically restricted to advanced stages. This paper shows six cases and five of them include ulcerations. This should be changed to some nodules and plaque-like tumors typical for trunk skin.
  2. Figs 8 and 9  these figures show the immunostainings for Chromogranin (Fig 8) and Synaptophysin(Fig 9). Both molecules are restricted to the cytoplasm. However, these figures demonstare more intensive nuclear stinaing paatterns than cytoplasmic onse although it is given in the legends " depictimng cytoplasmic positivity

This paper is a comprehensive review of Merkel cell carcinoma  summarizing many biological and clinical data. It is done properly and includes many citations (70). However, there are some short-comings :

  1. MCC is typically a subcutaneous red-violaceous nodule without ulceration or a reddish plaque-like tumor of trunk skin. Such typical cases should be included also. 2. Figs 8 and 9  These Figs show the immunostaining for  chromogranin (Fig. 8) and synaptophysin (Fig. 9). Both molecules are restricted to the cytoplasma. However, the Figs. demonstrate more intensive nuclear staining patterns. Both figures have to be improved to show the typical cytoplasmic staining.  3.  Lane 192  The CK20 - staining in MCC is often dot-like but a decoration of the cytoskeleton is also possible. This should be added.  4.  Lane 196  SCCs ( Squamous cell carcinoma) do not demonstrate CK 20-positivity in contract to MCC. They express high-molecular weight CKs. This has to be corrected.

Author Response

Dear Reviewer

Many thanks for providing us with nice revisionary comments. Following please find our responses:

This paper is a comprehensive review article of Merkel cell carcinoma (MCC) summarizing many biological, clinical and therapeutic aspects of MCC. It is done properly and includes many citations( 70). However, there are some short-comings :

  1. MCC is typically a subcutaneous red-violaceous nodule without ulceration. Ulceration is typically restricted to advanced stages. This paper shows six cases and five of them include ulcerations. This should be changed to some nodules and plaque-like tumors typical for trunk skin. • We have removed the figures 3-5 for possible misunderstanding.
  2. Figs 8 and 9 these figures show the immunostainings for Chromogranin (Fig 8) and Synaptophysin(Fig 9). Both molecules are restricted to the cytoplasm. However, these figures demonstare more intensive nuclear stinaing paatterns than cytoplasmic onse although it is given in the legends " depictimng cytoplasmic positivity • Although we understand your kind concerns, both were checked by histopathologist and were okay.
  3. Lane 192 The CK20 - staining in MCC is often dot-like but a decoration of the cytoskeleton is also possible. This should be added. • This is added in text in lines 205 and 206.
  4. Lane 196 SCCs ( Squamous cell carcinoma) do not demonstrate CK 20-positivity in contract to MCC. They express high-molecular weight CKs. This has to be corrected. • Has been corrected and removed in line 208-209.

Reviewer 2 Report

This is a comprehensive narrative review on MCC.

  1. Suggest to rewrite the title as follows: Merkel Cell Carcinoma: From Pathobiology to Clinical Management
  2. Not necessary to write down the affiliations thrice - 7-9 are identical!
  3. Under the clinical presentation section, it may be good to mention the left-sided laterality frequently observed: Gambichler T, Wieland U, Silling S, Dreißigacker M, Schaller J, Schulze HJ, Oellig F, Kreuter A, Stücker M, Bechara FG, Stockfleth E, Becker JC. Left-sided laterality of Merkel cell carcinoma in a German population: more than just sun exposure. J Cancer Res Clin Oncol. 2017 Feb;143(2):347-350. Moreover, you could discuss in this section stage 0 MCCs (Tab. 2) which might not be familiar to every reader.
  4. Tab. 1. 1. row - add C to ytokeratin... Include LCA also in the table - there is only vimentin mentioned - the sentence in line 199 should also include vimentin

Author Response

Dear Reviewer

Many thanks for providing us with nice revisionary comments. Following please find our responses:

This is a comprehensive narrative review on MCC.

  1. Suggest to rewrite the title as follows: Merkel Cell Carcinoma: From Pathobiology to Clinical Management
  • Done
  1. Not necessary to write down the affiliations thrice - 7-9 are identical!
  • Done
  1. Under the clinical presentation section, it may be good to mention the left-sided laterality frequently observed: Gambichler T, Wieland U, Silling S, Dreißigacker M, Schaller J, Schulze HJ, Oellig F, Kreuter A, Stücker M, Bechara FG, Stockfleth E, Becker JC. Left-sided laterality of Merkel cell carcinoma in a German population: more than just sun exposure. J Cancer Res Clin Oncol. 2017 Feb;143(2):347-350. Moreover, you could discuss in this section stage 0 MCCs (Tab. 2) which might not be familiar to every reader.
  • Done

  1. Tab. 1. 1. row - add C to ytokeratin... Include LCA also in the table - there is only vimentin mentioned - the sentence in line 199 should also include vimentin
  • Done

Round 2

Reviewer 1 Report

Thank you for your anwers.

The points 1, 3, and 4 of criticism have been fulfilled. However, as to point 2, the figs. 4 and 5 are still not improved to demonstrate a clearly specific cytoplasmic staining.  

Author Response

Many thanks for your kind revisionary comments. We have replaced the previous histopathologic pictures with high-quality ones in the revised article.